# Gut Butyrate Reduction in Blood Pressure Is Associated with Other Vegetables, Whole Fruit, Total Grains, and Sodium Intake

**DOI:** 10.3390/nu17081392

**Published:** 2025-04-21

**Authors:** Lauren San Diego, Taylor Hogue, Jarrad Hampton-Marcell, Ian M. Carroll, Troy Purdom, Heather Colleran, Marc D. Cook

**Affiliations:** 1Division of Nutritional Sciences, Cornell University, Ithaca, NY 14850, USA; lds224@cornell.edu; 2Department of Kinesiology, North Carolina Agriculture and Technical State University, Greensboro, NC 27411, USA; thogue@aggies.ncat.edu (T.H.); tpurdom@ncat.edu (T.P.); 3Department of Biological Sciences, University of Illinois Chicago, Chicago, IL 60607, USA; marcell2@uic.edu; 4Department of Nutrition, University of North Carolina at Chapel Hill, Chapel Hill, NC 27599, USA; ian_carroll@med.unc.edu; 5Exercise Physiology Integration and Collaboration (EPIC) Laboratory, North Carolina Agricultural and Technical State University, Greensboro, NC 27411, USA; hlcolleran@uncg.edu; 6Office of Research and Engagement, University of North Carolina at Greensboro, Chapel Hill, NC 27599, USA; 7Center for Integrative Health Disparity and Equity Research, College of Health & Human Sciences, North Carolina Agricultural and Technical State University, Greensboro, NC 27411, USA

**Keywords:** gut butyrate, diet quality, nutrients, hypertension, African Americans

## Abstract

Background: African Americans (AA) are disproportionally affected by hypertension (HTN). Gut microbiome metabolites (e.g., butyrate) may mediate the relationship between the microbiome and blood pressure (BP). Previous research reports a consistent indirect relationship between gut butyrate, a product of gut microbial nutrient fermentation, and BP. Thus, this study assessed the relationship between individual diet intake on BP changes after a butyrate treatment. Methods: AA aged 30–50 with HTN underwent treatment with a blinded placebo (5 mmol) and butyrate enema (80 mmol) with a one-week washout period. Ambulatory BP monitors collected measures up to 24 h post-enema. The Nutrition Data System for Research was used to assess diet and Healthy Eating Index (HEI-2015) scores from diet records. Paired *t*-tests and Kendall’s correlation tests determined group differences and relationships between variables (*p* < 0.05). Results: Positive correlations were found between other vegetables and 24 h diastolic BP (r = 0.64), daytime diastolic BP (r = 0.68), and MAP (r = 0.72). Positive correlations were also found between 24 h systolic BP and HEI-2015 greens and beans sub-scores (r = 0.64) and 24 h DBP and total vegetables (r = 0.64). Negative correlations were found between nighttime arterial stiffness and total grain intake (r = −0.71). Conclusion: These data suggest diet impacts BP measures in response to acutely increasing gut butyrate. These results provide preliminary evidence linking food groups, not individual nutrients, with BP outcomes and gut butyrate availability.

## 1. Introduction

Within the United States (US), hypertension (HTN) affects over 116 million people, nearly half of the adult population; less than a quarter of those with the diagnosis have managed HTN [1]. HTN increases the risk for heart disease and stroke, some of the leading causes of death in the US, and has been attributed with more than 670,000 deaths in 2020 [2]. Additionally, HTN has been trending upward since 2013–2014 [3]. Over 56% of African Americans (AA) are estimated to have HTN, which is disproportionately higher than the 48.1% prevalence in the total population [1]. Because AA people have been underrepresented in HTN studies [4], there is a critical need to understand potential causes and effective prevention and treatment strategies for this population.

Recent data have highlighted an inverse association between HTN and gut microbial butyrate production [5,6]. Butyrate, a short-chain fatty acid (SCFA), is produced by gut microbes and a key metabolite that can lower blood pressure (BP) [7,8]. A cross-sectional analysis found inverse associations between BP and both serum and fecal butyrate in overweight/obese cancer survivors, suggesting the potential of increasing blood or gut butyrate as an intervention for HTN [9]. However, an increase in plasma butyrate resulting from daily oral administration of sodium butyrate for four weeks was found to increase BP in Dutch participants with stage I HTN [10]. In contrast, differences in usual dietary patterns across regions and countries are needed to understand these conflicting reports. Further, the mechanisms associated with the processing and absorption of oral butyrate consumption need further study when compared to how butyrate is normally produced by gut microbes in the colon.

Diet is a contributor to gut microbial diversity and SCFA production. However, there remains a gap in knowledge on how dietary patterns influence gut butyrate production and mediation of vascular pathologies in this group, considering that diet can be examined in many ways, such as nutrients, food groups, and diet quality. Few studies connect diet, BP, and butyrate, and those studies only assessed individual nutrients or food groups [11,12,13,14], and reports vary in the measures of butyrate (i.e., measurements in serum or fecal samples). We found one study that examined the total diet [15], while no reports were found to assess diet quality. For example, one study using a mouse model found that a high-sodium diet had no significant effects on fecal butyrate [11]. Another study using a mouse model found that butyrate added to drinking water may blunt BP increases resulting from low-fiber diets [12]. Two studies examined diet and butyrate in humans with limited consideration of BP measures [13,15]. They found that in addition to reducing BP, reduced-sodium diets (2000 mg/d) and plant-based protein-enriched Dietary Approaches to Stop Hypertension (DASH) diets (48% carbohydrates, 25% protein, 27% fat) also increased serum butyrate, which is promising, but still limited in the scope of participants studied. In all, there is a paucity of literature connecting gut butyrate with diet and BP outcomes, specifically in populations with the greatest burden of HTN.

The purpose of this study was to examine the influence of normal dietary patterns on 24 h BP responses to an acute increase in gut butyrate in sedentary AA adults with mild HTN. As butyrate is a product of gut microbial fermentation of dietary fiber [6,7,8,16], usual diet intake was assessed to quantify participants’ diet quality and fiber consumption. It was hypothesized that elevated BP would be associated with low diet quality (including low fiber consumption). Therefore, the authors expected inverse relationships between BP and diet as measured by the Healthy Eating Index (HEI-2015), fiber, fruits, vegetables and whole grains. Additionally, we expected to observe direct relationships between sodium and saturated fat in this small cohort of HTN and non-HTN participants. To test if butyrate abundance in the gut can regulate BP, it was predicted that individuals with HTN and poorer diet quality would experience a significant BP reduction to an acute increase in gut butyrate. We also expected to find direct relationships between BP changes within 24 h after treatment and diet measures linked with higher fiber intake, while expecting inverse relationships with sodium and saturated fat.

## 2. Materials and Methods

### 2.1. Study Population

Participants were recruited via flyers from primary care and community settings surrounding a historically Black college/university. Prospective participants were included if they were aged 30–50, had normal BP (non-HTN; SBP = 90–129 mmHg and DBP = 60–89 mmHg) or stage I–II HTN (SBP = 130–159 mmHg and diastolic blood pressure, DBP = 80–99 mmHg), self-identified as AA, were sedentary, and had written clearance from a primary care physician for participation. Participants aged 30–50-year-old were targeted because they are more likely to have early-stage-I HTN and less likely to be taking BP medication. Therefore, we could assess factors such as diet associated with early stages of HTN. Participants were considered sedentary if they engaged in regular aerobic exercise 2 times or less per week for less than 20 min per session and had a sedentary profession. Participants were excluded if they had a diagnosis of heart, kidney, liver, or lung disease; peripheral vascular disease; diabetes mellitus; inflammatory diseases (inflammatory bowel disease, rheumatoid arthritis, and systemic lupus erythematosus); cancer within the past six months; kidney stones or gallbladder problems; uncontrolled HTN; history of heart attack; taken medications related to HTN or metabolic function within the past month; or were currently smoking.

### 2.2. Study Design

This study utilized a double-blind, crossover randomized control trial (RCT) where each HTN participant was their own control for the intervention. The control group for this study consisted of those in the non-HTN group (no intervention). Their anthropometric and body composition measurements, diet logs, blood draws, and ambulatory blood pressure monitor (ABPM) data were used as comparisons to the age- and sex-matched participants in the intervention group baseline measures. After completing the blood draw, control group participants returned home wearing the ABPM. After 24 h, they returned the ABPM to the laboratory.

Participants with HTN were randomized to self-administer an enema of sodium butyrate (80 mM butyrate in 0.9% saline, 60 mL total) or control saline (5 mM butyrate in 0.9%, 60 mL total). Randomization of the enemas was completed by an individual not associated with the study by choosing the letter associated with an enema (A or B) from an envelope. All subjects received the randomly chosen enema as their first enema for visit 1. After a washout-period of 7 days, participants with HTN returned for visit 2 to self-administer the enema opposite of their first dose and followed the same protocol. The compounding pharmacist blinded researchers and provided an envelope with the concentration of enema (A and B) at the end of the study. The use of the 80 mM butyrate enema has been reported safe in healthy volunteers and clinical populations [17,18]. Butyrate has an odor, so a 5 mM butyrate enema was used as the control to keep the participants and researchers blinded. Participants received verbal and written instructions on how to self-administer the enema and were instructed to consume the same foods recorded on their three-day written diet records.

### 2.3. Blood Pressure and Body Composition Measures

Manual resting BP measurements taken in the supine position were collected 3 times within 3 min on the non-dominant arm after lying down 5 min at the initial screening visit. The American Heart Association guidelines were used to categorize participants as HTN or non-HTN [19]. Height, weight, body composition (% fat mass, waist circumference (WC), waist-to-hip ration (WHR) were also measured using the bioelectric impedance scale Body Composition Analyzer 514 (SECA; Chino, CA, USA). Resting and total energy expenditure were calculated from body composition measures obtained from bioelectric impedance test.

Participants were fitted with a 24 h ABPM (Mobil-O-Graph^®^ 24 h PWA (IEM) ambulatory blood pressure monitor device), a non-invasive method of measuring BP and arterial stiffness. At the butyrate enema intervention visits 1 and 2, the monitor was worn on the non-dominant arm for 24 h, recording BP every half hour during waking hours and every hour during sleeping hours and returned the next day. If a BP reading was missed, the ABPM took another reading within 3 min up to 3 times to minimize the loss of data points. Greater than 85% of measurements were obtained. For each 24 h BP measurement, HR, SBP, DBP, MAP, and stiffness (measured via Augmentation Index, Aix, and Post Wave Velocity, PWV) were calculated by the Mobil-O-Graph unit, which has been validated to gold standard measures of applanation tonometry [20,21].

### 2.4. Diet Measures

At their first baseline visit, participants were instructed on how to complete their 3-day diet records and given detailed examples. Three-day diet records were completed three days prior to data collection for both groups to assess normal dietary patterns that may be associated with BP status. In attempt to standardize normal diet consumption prior to the experimental visits HTN participants were given their diet records back and were instructed to consume the same foods 3 days before their experimental visits. A review of the recalls was performed in-person for HTN group during the 1 h post-enema wait time. These records were entered into the Nutrition Data System for Research 2021 (NDSR, University of Minnesota). Diet quality was measured using the Healthy Eating Index (HEI), a composite score that is based on the Dietary Guidelines for Americans (DGAs), with sub-scores on food groups and nutrient intake per 1000 kcals consumed. Higher scores denote better adherence to the DGAs [22]. The latest HEI scoring metric was based off the 2015–2020 DGAs (HEI-2015). We expected that total diet quality measured by the total HEI-2015 scores and intake of the fruit, vegetable, and whole grain food groups would be related to greater fiber intake. Therefore, poorer diets would be interpreted as lower HEI scores, lower intake of fruits, vegetables, and whole grains. Additionally, poorer diets would include higher sodium and saturated fat intake.

### 2.5. Statistical Analysis

As this was the first study of its kind, there was not data available to estimate the effect size of the proposed intervention (cross-sectional or cross-over study) on BP. Therefore, an *n* = 10 in each group was chosen, and these data will be used to determine appropriate ranges of effect sizes for future studies. Data were analyzed using RStudio (v.4.2.2; R Core Team, 2022) and presented as mean ± standard deviation (SD). Welsh unpaired *t*-tests determined significant differences between the HTN and non-HTN groups. Paired *t*-tests were used between butyrate treatment doses. Kendall’s tau nonparametric correlation tests determined significant correlations between variables. Significance was set at *p* < 0.05.

## 3. Results

### 3.1. Group Baseline Measurements

There were no significant differences between age, weight, or waist-to-hip ratio between the HTN and non-HTN group (Table 1). However, those in the non-HTN group had lower resting energy expenditure and total energy expenditure (TEE) than the HTN group (calculated from the body composition analysis). The HTN group had significantly higher resting SBP, resting DBP, and daytime DBP than the non-HTN group (Table 1). Although there were significant differences in resting BP, 24 h ambulatory BP measures (other than daytime DBP) were not different between groups. There were also no significant group differences observed between measures of HR and arterial stiffness (i.e., AIx or PWV).

### 3.2. Diet Intake Measures

The HTN group had significantly higher intake of energy (kcals), saturated fat, total fat per kg of body weight, sodium (Na) to potassium (K) ratio, and no differences in overall fiber and butyrate intake (Table 2). The HTN group also had almost double the intake of total grains and refined grains compared to the non-HTN group (Table 2). There were no differences between groups in whole grain intake. The HTN group also consumed significantly less seafood than the non-HTN group. Diet quality was similar between the groups, and those with HTN had significantly better HEI-2015 sodium sub-scores than those without HTN (Table 3).

### 3.3. Correlations in Diet Measures and Associations with BP Changes After Butyrate Treatment

For the usual diet measures assessed, both resting SBP, 24 h SBP, and daytime SBP were moderately and significantly (*p* < 0.05) correlated to dietary butyrate intake (r = 0.48; r = 0.45; and r = 0.45, respectively). Resting SBP was also moderately and significantly correlated to total energy intake (r = 0.59), saturated fat intake (r = 0.60), and soluble fiber intake (r = 0.45). For the participants in the HTN group who underwent the butyrate treatment, changes in BP measures between the two treatments (5 mM vs. 80 mM) were significantly correlated (moderately high) to the intake of total vegetables, other vegetables (e.g., broccoli, cabbage, and onions), total grains, and sodium, HEI-2015 greens and beans sub-scores, and whole fruits sub-score. Changes in 24 h DBP, daytime DBP, and MAP were significantly correlated to the other vegetables food group (r = 0.64; r = 0.68; and r = 0.72, respectively). Changes in daytime HR had a significantly negative correlation to the HEI-2015 whole fruits sub-score (r = −0.71). Changes in nighttime HR also had a significant negative correlation to sodium intake (r = −0.74). Finally, changes in nighttime AIx had a significant negative correlation to total grain intake (r = −0.71). All correlations are represented in Table 4.

## 4. Discussion

The purpose of this study was to explore the association of normal diet patterns on BP responses to a butyrate enema in sedentary AA adults at stage I–II HTN. This study found that BP changes in response to the butyrate enema were significantly correlated with diet measures related to the “other vegetables” group, total grains, and whole fruit. As lower gut butyrate production has been associated with greater BP, assessing diet patterns is crucial in the evaluation of individuals’ capacity to produce gut butyrate and its impact on acute BP changes in our intervention. Direct administration of butyrate to the gut resulted in statistically significant improvements to daytime SBP and MAP. Lower consumption of other vegetables (e.g., broccoli, cabbage, and onions) was significantly associated with greater decreases in daytime DBP and MAP. Other vegetables include all other fresh, frozen, canned, cooked, or raw vegetables, such as beets, cabbage, cauliflower, cucumber, green beans, onions, okra, and squash [23]. Additionally, greater intake of dietary butyrate and soluble fiber were significantly correlated with higher SBP among both HTN and non-HTN participants.

### 4.1. Diet Quality Measures

Contrary to what authors expected, there were no statistically significant differences in total diet quality between the HTN and non-HTN group as measured by the HEI-2015, though there were group differences in nutrient intake. Notably, the scores observed in the HTN group in this study (42.6 ± 12.5) were even lower than those observed in other studies with AA [24,25,26]. Here, the HTN group had a better average HEI-2015 sub-score for sodium than the non-HTN group. This may represent differences in access to food sources and/or efforts by the HTN group to improve their diet given their previous knowledge of having an elevated BP.

### 4.2. Correlations with Normal Diet Measures

The moderate and significant correlation found between dietary butyrate and SBP in humans is a novel finding from this study. The positive correlations suggest that higher dietary butyrate intake was associated with higher SBP, in contrast to previous animal studies that found addition of sodium butyrate to diet decreased BP [12,14,27]. However, one study found that the addition of butyrate to a high-fat diet (20.0% kcal protein, 35.0% carbohydrates, and 45.0% kcal fat) did not significantly change SBP after 9 and 12 months in a HTN model of LDLr −/− mice [28]. Butyrate is a short-chain saturated fatty acid, so dietary butyrate comes from foods with saturated fat, which have been linked to cardiovascular risk factors, such as increased BP and adverse outcomes [29,30,31]. Though there were no significant differences in butyrate intake by HTN status, there were significant differences in saturated fat intake. Therefore, the significant positive correlations between dietary butyrate and SBP from this study are aligned with DGA and DASH diet recommendations to limit saturated fat intake to reduce cardiovascular risk.

The significant positive correlation found between SBP and soluble fiber was unexpected, as previous meta-analyses and a Cochrane review found inverse relationships between BP and fiber intake [32,33,34]. Within this study’s sample, those with HTN consumed more energy, total grains, and refined grains than those without HTN. Therefore, this significant relationship with fiber may have reflected outcomes related to total and refined grain intake differences, despite insignificant differences in total fiber by HTN status. Also unexpected was the lack of statistically significant correlations between BP measures and total HEI-2015 scores, and consumption of fruits, vegetables, and whole grains. However, this may be a reflection on the lack of statistically significant differences between the HTN and non-HTN participants in these diet measures.

### 4.3. Correlations to BP Changes from the Butyrate Treatment

It was hypothesized that the butyrate treatment would result in greater BP improvement for those with lower diet quality. Therefore, it was expected that direct relationships would be observed between BP and diet measures associated with improving cardiac health, such as greater fiber, fruits, vegetables, and whole grains intake. Inverse relationships were expected between BP and diet measures associated with worsening cardiac health such as sodium, and saturated fat. The data were consistent with the expectations for sodium intake and several vegetable measures: total vegetables, greens and beans, and the category of other vegetables (e.g., broccoli, cabbage, and onions).

Intake of other vegetables was significantly and positively correlated to 24 h DBP, daytime DBP and daytime MAP, suggesting that those with the lowest intake of other vegetables experienced the greatest decreases in these measures after the high-dose butyrate enema as compared to those who consumed more other vegetables. The significant findings between 24 h SBP and DBP with total vegetable intake and HEI-2015 greens and beans sub-scores provide additional support to the relationship between vegetables and these BP measures. Together, these observations provide more evidence concerning the importance of vegetable intake on the gut’s participation in vascular health. Additionally, those with higher intake of vegetables may not have had a notable decrease in daytime DBP and MAP, perhaps because the fiber provided by their vegetable intake was sufficient for endogenous butyrate production via the microbiome. Therefore, additional butyrate provided by the enema may have had no benefit to these BP measures.

The moderately high significant negative correlation of total grains and AIx suggests that those with higher grains consumption responded to the treatment with greater improvements in arterial stiffness during the nighttime hours. The findings of this study build on the limited studies that report on relationships between grain consumption and AIx [35,36]. While it was previously reported that diet patterns with high refined grain and low whole grain consumption during late-night eating were associated with higher AIx [35], this current study demonstrated the benefits of gut butyrate on this measure of arterial stiffness for those who ate more grains. Together these findings suggest that diet works synergistically with the effects of butyrate abundance in the gut to improve BP at early stage (I) HTN, when lifestyle modifications (e.g., diet) may be most beneficial.

### 4.4. Limitations

As this was a proof-of-concept study to examine the impact of acutely increasing gut butyrate on BP, the small sample size limits the ability to detect other significant correlations of BP measures to the butyrate enema treatment and diet. We attempted to address diet variability during the 7-day washout period between visits by asking them to consume the same foods at least 3 days before and providing a copy of their initial diet recall to follow. One limitation to this is that it was only self-reported that they adhered. Given the small sample, the outliers were not removed, and a linear relationship between diet variables and BP measures was not assumed. The use of Kendall’s correlation co-efficient is helpful as it is difficult to determine if the outliers in this study are true outliers to this population.

Additionally, the findings of this study are limited only to AA and may not be applicable to other racial groups, especially when diet quality in this population has been documented to be lower than other racial/ethnic groups [26,37,38,39,40,41,42,43,44,45,46,47]. However, the previous literature examining fecal butyrate found no racial differences between AA and White [48,49] or Hispanic populations [48]. Furthermore, previous studies that examined the effect of diet changes on serum butyrate found no significant interactions with race [13,15]. Future analyses of these data should include fecal butyrate to provide a more comprehensive understanding of gut butyrate’s role in HTN within AA.

### 4.5. Strengths

There are several strengths to this study. The inclusion of nutrients, food groups, and diet quality is a robust method to examine different aspects of usual diet in participants. The correlations to food groups and HEI-2015 scores may provide practical recommendations in translating findings to diet intervention studies. Additionally, the use of ABPM provides a comprehensive assessment of the BP outcomes resulting from the butyrate treatment.

This study reports a few novel findings. Dietary butyrate intake was positively correlated to baseline SBP, which suggests that butyrate from the gut, not diet, may be responsible for BP reductions. Lower intake of vegetables was significantly associated with greater BP decreases resulting from direct gut butyrate administration. Additionally, findings from this study report significant correlations relating diet with 24 h BP measures, such as arterial stiffness (AIx), which are not always reported in cardiovascular and nutrition research in this group. These results start to fill some gaps in the literature linking diet to measures of vascular health.

Finally, this study provides valuable insight concerning usual diet influences on BP and vascular responses to a direct increase in gut butyrate in a group that is disproportionally affected by HTN. The majority of studies using butyrate as an intervention in the context of HTN were on animal models [12,14,50,51,52,53]. The few human studies relating butyrate to vascular outcomes that measured circulating butyrate demonstrated that diets lower in sodium [13] and DASH diets higher in protein [15] increased circulating butyrate while improving BP. However, these studies did not consider participants’ normal diet intake. This current study complements the literature by demonstrating in AA participants’ usual diet that higher dietary intake of butyrate was significantly associated with higher SBP and that greater decreases in SBP, MAP, HR, and AIx from gut butyrate were significantly associated with a lower intake of other vegetables (e.g., broccoli, cabbage, and onions) and a higher intake of sodium.

## 5. Conclusions

These findings contribute to the research involving the guts relationship to BP status by isolating the effect of a product of microbiome fermentation, butyrate, to diet. The lower consumption of total vegetables, other vegetables (e.g., broccoli, cabbage, and onions), and greens and beans was associated with greater improvements to BP in response to acute increases in gut butyrate. In contrast, greater servings of total grains were associated with greater improvements to arterial stiffness in response to acute increases in gut butyrate. These results provide evidence that diet works synergistically with gut butyrate to improve vascular health, and are consistent with previous studies connecting food groups, instead of individual nutrients, to improved cardiovascular outcomes.

## Figures and Tables

**Table 1 nutrients-17-01392-t001:** Baseline characteristics and vascular measures of participants.

	Normotensive (*n* = 10)	Hypertensive (*n* = 10)	*p*-Value ^a^
Age	45.6 (9.6)	40.8 (9.5)	0.2756
Weight (kg)	85.9 (18.5)	99.7 (23.0)	0.1568
BMI ^b^	30.2 (4.9)	35.4 (7.6)	0.0837
% Fat Mass	38.0 (6.9)	38.7 (11.6)	0.8628
WC ^c^	36.8 (5.6)	41.1 (7.9)	0.1726
Waist-to-Hip Ratio	0.85 (0.09)	0.91 (0.08)	0.1281
REE ^d^	1679 (326)	1830 (206)	<0.0001
TEE ^e^	2850 (560)	3037 (384)	<0.0001
Resting SBP ^f^	118 (4.9)	135 (6.1)	<0.0001
Resting DBP ^g^	74 (3.5)	84 (9.0)	0.0078
24 h SBP ^f^	124 (14.6)	133 (12.3)	0.1268
24 h DBP ^g^	77 (7.1)	85 (9.4)	0.0535
Day SBP ^f^	128 (16.3)	138 (13.5)	0.1820
Day DBP ^g^	80 (7.4)	89 (9.7)	0.0313
Night SBP ^f^	114 (12.9)	123 (13.6)	0.1370
Night DBP ^g^	69 (7.8)	73 (8.4)	0.2223
Day HR ^h^	74 (14.6)	80 (8.5)	0.2558
Night HR ^h^	66 (13.7)	74 (12.4)	0.1991
Day MAP ^i^	102 (9.5)	111 (10.6)	0.0531
Night MAP ^i^	89 (9.5)	96 (9.8)	0.1218
Day AIx ^j^	24.7 (6.7)	24.4 (6.3)	0.9194
Night AIx ^j^	21.5 (9.6)	17.4 (11.0)	0.3867
Day PWV ^k^	6.9 (1.2)	6.8 (1.3)	0.9006
Night PWV ^k^	6.4 (1.0)	6.4 (0.9)	0.9806

Data reported as mean (SD). ^a^ Between hypertension status (totals). ^b^ BMI: body mass index, values reported as kg/m^2^. ^c^ WC: waist circumference, values reported as inches. ^d^ REE: resting energy expenditure, values reported as kcals. ^e^ TEE: total energy expenditure, values reported as kcals. ^f^ SBP: systolic blood pressure, values reported as mmHg. ^g^ DBP: diastolic blood pressure, values reported as mmHg. ^h^ HR: heart rate, values reported as beats per minute. ^i^ MAP: mean arterial pressure, values reported as mmHg. ^j^ AIx: augmentation index, values reported as %. ^k^ PWV: post-wave velocity, values reported as m/s.

**Table 2 nutrients-17-01392-t002:** Average daily nutrient intake and food group servings of normotensive and hypertensive groups.

	Normotensive (*n* = 10)	Hypertensive (*n* = 10)	*p*-Value ^a^
Total Energy (kcals)	1646 (468)	2687 (734)	0.0018
% Carbohydrates	42.7 (16.2)	43.0 (7.8)	0.9576
% Fat	37.1 (10.9)	43.1 (5.9)	0.1516
% Protein	20.0 (6.6)	15.4 (2.9)	0.0610
Kcals/kg	8.8 (2.3)	12.8 (4.6)	0.0293
g Carbohydrates/kg	1.0 (0.4)	1.4 (0.6)	0.0816
g Fat/kg	0.4 (0.1)	0.6 (0.2)	0.0095
g Protein/kg	0.4 (0.1)	0.5 (0.2)	0.3634
Saturated fat (g)	21.6 (10.0)	43.2 (20.3)	0.0096
Fiber (g)	13.1 (7.9)	17.7 (5.8)	0.1564
Soluble Fiber (g)	3.3 (2.2)	5.4 (2.6)	0.0646
Insoluble Fiber (g)	9.7 (5.6)	12.1 (4.2)	0.2866
Sodium (mg)	3902 (1358)	4494 (1258)	0.3253
Potassium (mg)	2051 (625)	2297 (601)	0.3812
Na:K ^b^ ratio	2.0 (0.8)	2.0 (0.6)	0.0018
Dietary Butyrate (g)	0.3 (0.3)	0.4 (0.3)	0.1516
Total Vegetables ^c^	3.2 (1.7)	2.4 (1.3)	0.2569
Dark Green Vegetables ^c^	1.0 (1.3)	0.5 (0.6)	0.2552
Red and Orange Vegetables ^c^	0.6 (0.9)	0.4 (0.3)	0.5401
Beans, Peas, Lentils ^c^	0.2 (0.4)	0.1 (0.1)	0.2540
Starchy Vegetables ^c^	0.4 (0.6)	0.5 (0.4)	0.8534
Other Vegetables ^c^	1.0 (1.0)	1.0 (0.8)	0.9461
Fruits ^c^	1.0 (0.9)	0.8 (0.7)	0.6393
Total Grains ^c^	4.6 (2.8)	9.1 (5.1)	0.0278
Whole Grains ^c^	0.7 (0.9)	1.3 (1.3)	0.2880
Refined Grains ^c^	3.9 (2.4)	7.8 (4.8)	0.0365
Dairy ^c^	0.8 (0.5)	1.1 (0.9)	0.2583
Total Protein Foods ^c^	8.2 (4.8)	9.0 (3.2)	0.6819
Meats, Poultry, Eggs ^c^	5.9 (4.0)	8.5 (3.0)	0.1175
Seafood ^c^	2.0 (2.7)	0.01 (0.03)	0.0430
Nuts, Seeds, Soy Products ^c^	0.3 (0.5)	0.5 (1.0)	0.6278

Data reported as daily mean (SD). ^a^ Between hypertension status (totals). ^b^ Na:K: sodium-to-potassium. ^c^ Food group reported as servings.

**Table 3 nutrients-17-01392-t003:** Average Healthy Eating Index 2015 scores for normotensive and hypertensive groups.

	Normotensive (*n* = 10)	Hypertensive (*n* = 10)	*p*-Value ^a^
Total Score	50.4 (15.6)	42.6 (12.5)	0.2308
Total Fruits	1.8 (1.6)	1.0 (0.7)	0.2026
Whole Fruits	2.1 (2.2)	1.5 (1.4)	0.5004
Total Vegetables	3.7 (1.5)	2.3 (1.6)	0.0619
Greens and Beans	3.6 (2.0)	2.1 (2.0)	0.1191
Whole Grains	2.3 (3.4)	2.7 (2.9)	0.7653
Dairy	3.4 (1.8)	2.9 (1.8)	0.5168
Total Protein	4.9 (0.4)	4.7 (0.6)	0.4471
Seafood and Plant Protein	3.1 (2.5)	1.2 (2.0)	0.0758
Fatty Acids	5.7 (3.2)	5.2 (4.1)	0.7351
Refined Grains	5.6 (3.6)	5.2 (4.1)	0.8287
Sodium	0.8 (1.6)	3.7 (3.1)	0.0217
Added Sugars	7.5 (3.2)	6.3 (3.8)	0.4634
Saturated Fats	6.1 (3.8)	3.8 (3.1)	0.1657

Data reported as mean (SD). ^a^ Between hypertension status (totals).

**Table 4 nutrients-17-01392-t004:** Significant correlations of diet measures and BP changes from the butyrate treatment.

BP Measure	Diet Measure	r ^a^	*p*-Value
Baseline Resting SBP ^b^	Total kcals	0.5858	0.0067
Baseline Resting SBP ^b^	Dietary Butyrate (g)	0.4762	0.0338
Baseline Resting SBP ^b^	Saturated Fat (g)	0.5963	0.0055
Baseline Resting SBP ^b^	Soluble Fiber (g)	0.4486	0.0473
Baseline 24 h SBP ^b^	Dietary Butyrate (g)	0.4468	0.04823
Baseline Day SBP ^b^	Dietary Butyrate (g)	0.4509	0.0460
Change in 24 h SBP ^b^	HEI-2015 ^g^ Greens and Beans	0.6437	0.0446
Change in 24 h SBP ^b^	Total Vegetables (servings)	0.6441	0.0445
Change in 24 h SBP ^b^	Other Vegetables (servings)	0.6441	0.0445
Change in Day DBP ^c^	Other Vegetables (servings)	0.6751	0.0322
Change in Day MAP ^d^	Other Vegetables (servings)	0.7191	0.0191
Change in Night AIx ^e^	Total Grains (servings)	−0.7143	0.0465

^a^ Kendall’s tau correlation coefficient. ^b^ SBP: systolic blood pressure. ^c^ DBP: diastolic blood pressure. ^d^ MAP: mean arterial pressure. ^e^ AIx: augmentation index. ^g^ HEI-2015: Healthy Eating Index 2015.

## Data Availability

De-identified data can be made available upon request to the corresponding author. Clinical data are available at clinicaltrials.gov NCT04415333.

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
