# Peer review of "Gut Butyrate Reduction in Blood Pressure Is Associated with Other Vegetables, Whole Fruit, Total Grains, and Sodium Intake"

_nutrients, 2025, doi:10.3390/nu17081392_

Round 1

Reviewer 1 Report

Comments and Suggestions for Authors

Interesting paper and findings, but could benefit from some reorganization and improvements in clarity. Specific comments:

Title: The title is confusing, mainly the beginning part: "Acute increase in gut butyrate reduction in blood pressure is..." Maybe reword to something like "Gut butyrate reduction in blood pressure is associated with..." because when the title has both increase and reduction it can obscure which things are increasing and reducing. 

Abstract: In the third sentence, it reads like results rather than intro. Maybe write something like "In prior research, acute increases in gut butyrate via enema decrease daytime BP and mean arterial pressure (MAP) in AA with HTN" to demonstrate that this is already known before this study. The next sentence could be rewritten to "The objective of this study was to assess the relationship between individual diet intake on BP changes after gut butyrate treatment" so that this paper is differentiated from prior research in the sentence before. 

Introduction

  1. The sentence in line 50 about your proof-of-concept study is not cited, is that this paper or a prior paper? If it is a prior paper, please cite that here. If that research is part of this paper it should not be in the literature review section, and other research should be cited here.
  2. Make sure to mention that oral butyrate INCREASES blood pressure and be clear throughout the intro that gut butyrate and butyrate from oral intake have different effects. Can use the 2024 Verhaar study with doi: 10.1161/HYPERTENSIONAHA.123.22437 to talk about oral butyrate, and the Tilves study with doi: 10.1161/JAHA.121.024763 to talk about serum and gut butyrate. In line 58, when you write "BP-lowering effects of butyrate" make sure to write "gut butyrate", or "serum butyrate", same as in line 60 and 62. It may seem wordy, but make sure to specify when a study is writing about dietary, gut, or serum butyrate throughout because of the differing effects.

Methods

  1. Please put the methods section after the intro and not at the end of the paper, I was worried there was not a methods section.
  2. In line 285 it says at visit 1 and 2. What occurred during visit 1, and what occurred during visit 2? If the enema occurred at only one visit, what happened at the other visit?
  3. In lines 304-308, it is a little confusing. You write that they are instructed to eat the same foods 3 days before their experimental visits. Does that mean the first visit is just data collection and not experimental? Did you do anything to measure or ensure that the participants followed the same diet for a second 3 day period?
  4. The study design section would be better earlier in the methods section, right after study population, then followed by the measures sections and finally analysis. Many things in the measures section make more sense with the context provided in the study design section.
  5. The control group received no enemas? Are they solely compared to the HTN group at baseline?

Results

  1. Do not need to summarize the results at the top, save that for the discussion.
  2. The titles on tables 1 and 2 are kinda long. Table 1 could just be baseline characteristics of participants, don't need to write out study information. Also could combine tables 1 and 2, and say baseline characteristics and vascular measures, because 6 tables is a lot of tables.
  3. What is the scale on Table 4? Servings? Cups? Is this daily? Weekly? Is seafood measured in ounces? It is hard to interpret what the numbers mean, without indicating the scale like you did in Table 3 for each nutrient.
  4. Possibly combine tables 3 and 4 into nutrient and food group intake to reduce the number of tables. 

Discussion and Conclusion

  1. Please reorganize the conclusion to be right after the discussion (which should be the order if methods is put in before results).
  2. Please make sure the results summary that was in the results section is incorporated into the beginning of the discussion, if those sections reported different results.
  3. The conclusion could have a little more detail - instead of writing significant correlations to diet, can say which food groups were related to higher or lower BP measures. 

Author Response

We are grateful for the insightful and helpful reviewers comments that have assisted us in improving the clarity of the results we are reporting. We understand that this is time consuming so we appreciate it very much. Please find our responses to the comments below. 

REVIEWER 1

Interesting paper and findings but could benefit from some reorganization and improvements in clarity. Specific comments:

Title: The title is confusing, mainly the beginning part: "Acute increase in gut butyrate reduction in blood pressure is..." Maybe reword to something like "Gut butyrate reduction in blood pressure is associated with..." because when the title has both increase and reduction it can obscure which things are increasing and reducing. 

RESPONSE: Thank you for providing feedback to be more direct in the title that represents the finding. We have changed the title to: “Gut butyrate reduction in blood pressure is associated with Vegetables, Fruits, Grains, and Sodium Intake”.

Abstract: In the third sentence, it reads like results rather than intro. Maybe write something like "In prior research, acute increases in gut butyrate via enema decrease daytime BP and mean arterial pressure (MAP) in AA with HTN" to demonstrate that this is already known before this study. The next sentence could be rewritten to "The objective of this study was to assess the relationship between individual diet intake on BP changes after gut butyrate treatment" so that this paper is differentiated from prior research in the sentence before. 

 RESPONSE: Thank you. We have amended this sentence has been amended in line 20 (abstract).

Previous research reports a consistent indirect relationship between gut butyrate, a product of gut microbial nutrient fermentation, and BP. Thus, this study assessed the relationship between individual diet intake on BP changes after a butyrate treatment. Introduction

  1. The sentence in line 50 about your proof-of-concept study is not cited, is that this paper or a prior paper? If it is a prior paper, please cite that here. If that research is part of this paper it should not be in the literature review section, and other research should be cited here.

RESPONSE: Thank you for this comment. Our parent study to this report is the main result we are referring to here. It is currently under review in another journal (with minor revisions recently submitted) but it has taken a long time for review. We have removed the mention of our parent study in the introduction and have cited appropriate research to assist in explaining this phenomenon. Lines 50-62 have been amended.

  1. Make sure to mention that oral butyrate INCREASES blood pressure and be clear throughout the intro that gut butyrate and butyrate from oral intake have different effects. Can use the 2024 Verhaar study with doi: 10.1161/HYPERTENSIONAHA.123.22437 to talk about oral butyrate, and the Tilves study with doi: 10.1161/JAHA.121.024763 to talk about serum and gut butyrate. In line 58, when you write "BP-lowering effects of butyrate" make sure to write "gut butyrate", or "serum butyrate", same as in line 60 and 62. It may seem wordy, but make sure to specify when a study is writing about dietary, gut, or serum butyrate throughout because of the differing effects.

RESPONSE: Thank you for this comment. We completely understand and the sections in the introduction in (lines 50-59) has been updated and these key references are incorporated.  

Methods

  1. Please put the methods section after the intro and not at the end of the paper, I was worried there was not a methods section.

RESPONSE: Thank you for pointing this out. We followed the template provided but we have moved the methods section up. We hope the journal editors will accept this change also.  

  1. In line 285 it says at visit 1 and 2. What occurred during visit 1, and what occurred during visit 2? If the enema occurred at only one visit, what happened at the other visit?

RESPONSE: Thank you for this question. We have amended the methods section in lines 122-127 to improve the description and flow of the methods section. Visit 1 consisted of the first enema; Visit 2 consisted of the second enema as highlighted in lines 122-123.

  1. In lines 304-308, it is a little confusing. You write that they are instructed to eat the same foods 3 days before their experimental visits. Does that mean the first visit is just data collection and not experimental? Did you do anything to measure or ensure that the participants followed the same diet for a second 3 day period?

RESPONSE: Thank you for the opportunity to clarify. We hope that our amendment of the methods section (lines 122-127) provide clarity on the flow of the study. The first intervention visit (1) included administration of the first enema; the second intervention visit (2) included the administration of the second enema 7-days apart. A limitation of this study was that participants self-reported following the same diet prior to the second visit. But all participants were provided with a copy of their initial diet recall so they were aware of what foods they consumed. This has been added as a limitation section in line 318-319.

  1. The study design section would be better earlier in the methods section, right after study population, then followed by the measures sections and finally analysis. Many things in the measures section make more sense with the context provided in the study design section.

RESPONSE: Thank you for the opportunity to amend this. The study design has been moved accordingly.

  1. The control group received no enemas? Are they solely compared to the HTN group at baseline?

RESPONSE: This is correct; the control group received no enemas. They were solely a comparison group of AA participants that are normotensive. In the context of this small proof-of-concept study examining the impact of gut butyrate on its potential to lower BP, we would not expect BP to get any better than normal in the control group and the greatest impact would be in those with HTN. However, future studies could consider this intervention in both groups. Line 111-115 has been amended to indicate the control group’s measures were compared to the HTN’s baseline measures.

Results

  1. Do not need to summarize the results at the top, save that for the discussion.

RESPONSE: The first paragraph of the results have been removed.

  1. The titles on tables 1 and 2 are kinda long. Table 1 could just be baseline characteristics of participants, don't need to write out study information. Also could combine tables 1 and 2, and say baseline characteristics and vascular measures, because 6 tables is a lot of tables.

RESPONSE: Thank you for pointing this out. Tables 1 & 2 have been combined as requested.

  1. What is the scale on Table 4? Servings? Cups? Is this daily? Weekly? Is seafood measured in ounces? It is hard to interpret what the numbers mean, without indicating the scale like you did in Table 3 for each nutrient.

RESPONSE: Thank you for the opportunity to clarify. The title of Table 4 (now combined with Table 3 and renamed as Table 2) has been amended to indicate values reflect average servings per day of the listed food groups.

  1. Possibly combine tables 3 and 4 into nutrient and food group intake to reduce the number of tables. 
  1. RESPONSE: Thank you for pointing this out. Tables 3 & 4 have been combined as requested and renamed as Table 2.

Discussion and Conclusion

  1. Please reorganize the conclusion to be right after the discussion (which should be the order if methods is put in before results).

RESPONSE: Thank you for pointing this out. This has been amended.

  1. Please make sure the results summary that was in the results section is incorporated into the beginning of the discussion, if those sections reported different results.

RESPONSE: Thank you for the clarification; this has been incorporated at line 233-238.

  1. The conclusion could have a little more detail - instead of writing significant correlations to diet, can say which food groups were related to higher or lower BP measures. 

RESPONSE: Thank you for the clarification; this has been amended in line to highlight the findings related to responses from the treatment.

Reviewer 2 Report

Comments and Suggestions for Authors

Thank you for submitting the manuscript "Acute Increase in Gut Butyrate Reduction in Blood Pressure Is Associated with Other Vegetables, Whole Fruit, Total Grains, and Sodium Intake" to Nutrients. The article is well written and the research is interesting. I have a few minor considerations:

- I think the title doesn't sound right because it generates different interpretations such as (i) does the consumption of these foods alter gut butyrate?, (ii) is there no difference between the amount of vegetable intake, why is it important to include it in the title?, (ii) is there also a significant difference in the intake of refined grains, using this pattern to form the title, should it also be in the title?, and (iv) is the title confusing since when you read it for the first time it seems that the presence of dietary butyrate is beneficial? Consider changing the title to something more precise.

- Line #45: I think you need to include here what the percentage of the general population is.

- Introduction: it is important to add what the hypothesis of this study was. - Material and methods: must be presented in accordance with the standards of this journal.
- I believe that the best option would be to use international units, for example, for weight it is kg.
- All table titles must indicate what the values ​​are: averages? score? index? percentage? relative percentage?

Comments on the Quality of English Language

English Language is fine.

Author Response

We are grateful for the insightful and helpful reviewers comments that have assisted us in improving the clarity of the results we are reporting. We understand that this is time consuming so we appreciate it very much. Please find our responses to the comments below. 

REVIEWER 2

Thank you for submitting the manuscript "Acute Increase in Gut Butyrate Reduction in Blood Pressure Is Associated with Other Vegetables, Whole Fruit, Total Grains, and Sodium Intake" to Nutrients. The article is well written and the research is interesting. I have a few minor considerations:

- I think the title doesn't sound right because it generates different interpretations such as (i) does the consumption of these foods alter gut butyrate?, (ii) is there no difference between the amount of vegetable intake, why is it important to include it in the title?, (ii) is there also a significant difference in the intake of refined grains, using this pattern to form the title, should it also be in the title?, and (iv) is the title confusing since when you read it for the first time it seems that the presence of dietary butyrate is beneficial? Consider changing the title to something more precise.

              RESPONSE: Thank you for providing feedback to be more direct in the title that represents the finding. We have changed the title to: “Gut butyrate reduction in blood pressure is associated with Vegetables, Fruits, Grains, and Sodium Intake”.

- Line #45: I think you need to include here what the percentage of the general population is.

       RESPONSE: Thank you for this suggestion to provide context. Line 45 has been amended to include the %.

- Introduction: it is important to add what the hypothesis of this study was.

       RESPONSE: We apologize that this was not clear. The hypotheses and expected findings of this study is listed in lines 81-90.

- Material and methods: must be presented in accordance with the standards of this journal.

RESPONSE: Thank you for pointing this out. This has been amended.

- I believe that the best option would be to use international units, for example, for weight it is kg.

RESPONSE: Thank you for pointing this out. This has been amended in Table 1.  

- All table titles must indicate what the values ​​are: averages? score? index? percentage? relative percentage?

RESPONSE: Tables 3 & 4 have been amended to average intake of nutrients, food group servings, and health eating index scores. Table 5 diet measures have been amended to reflect the values of the diet measures.